# Patient Safety Incidents in Inpatient Psychiatric Settings: An Expert Opinion Survey

**DOI:** 10.3390/bs14111116

**Published:** 2024-11-20

**Authors:** Sophia Russotto, Andrea Conti, Kris Vanhaecht, José Joaquín Mira, Massimiliano Panella

**Affiliations:** 1Department of Translational Medicine, Università del Piemonte Orientale, 28100 Novara, Italy; russottosophia@gmail.com (S.R.); massimiliano.panella@med.uniupo.it (M.P.); 2Doctoral Program in Sports and Health—Patient Safety Line, Universitas Miguel Hernandez, 03202 Alicante, Spain; 3Doctoral Program in Food, Health, and Longevity, Università del Piemonte Orientale, 28100 Novara, Italy; 4Leuven Institute for Healthcare Policy, KU Leuven, 3000 Leuven, Belgium; kris.vanhaecht@kuleuven.be; 5Atenea Research, FISABIO, Hermanos López de Osaba, 03013 Alicante, Spain; jose.mira@umh.es

**Keywords:** psychiatric inpatient, patient safety, expert opinion, safety, clinical risk, risk management

## Abstract

Patient safety in psychiatric inpatient facilities remains under-researched despite its crucial importance. This study aims to address this gap by using expert opinion to estimate the frequency of diverse patient safety incidents (PSIs) in psychiatric settings and to compare it with the existing literature. Utilizing a seven-step approach, a questionnaire based on the World Health Organization’s International Classification for Patient Safety was developed and deployed. A total of 33 expert opinions were collected. Results showed a higher estimated incidence of PSIs in psychiatric settings compared to general healthcare, highlighting categories such as patient behavior, medication, and infrastructure as significant contributors. Experts emphasized the prevalence of incidents related to behavioral issues and inadequate infrastructure, areas often overlooked in the existing literature. Unlike general settings, psychiatric facilities appear more vulnerable to specific PSIs, such as those related to medication and building safety, underscoring the need for targeted safety measures. Our study suggests the existence of significant discrepancies between expert opinion and available research, with several underexplored domains in psychiatric patient safety.

## 1. Introduction

Patient safety, namely activities aimed to “lower risks, reduce avoidable harm, and minimize error impact” [1], has increasingly gained attention. Globally, over 3 million people die annually from unsafe care, with adverse events harming more than 10% of hospitalized patients. While extensively studied in general medical settings, psychiatric inpatient facilities face a significant gap in patient safety knowledge [2]. Indeed, these settings present unique safety challenges: patients’ vulnerable mental states, cognitive impairments, behavioral issues [3], extensive psychotropic medication use, and complex physical environments [4] critically influence patient safety.

Traditionally, research on patient safety has been based on data obtained from incident-reporting systems [5]. However, such systems suffer from an important reporting bias, which hampers their impact as a safety improvement tool [6,7]. Such limitations are even greater in psychiatry, with the majority of incidents not reported [8] due to the fear of consequences or the misrecognition of some adverse events as patient safety issues [9]. Moreover, incident-reporting systems may not capture the full spectrum of safety issues, particularly those that are normalized in the psychiatric inpatient setting, such as aggressions or behavior-related events [9].

Neglecting patient safety can critically impact both patients and staff. Risks include potential self-injury, aggression, and deteriorating mental health [10]. Insufficient safety measures may erode patient–staff trust and trigger the second victim phenomenon [11,12].

Given these limitations, expert opinion might be considered a valuable alternative source of information in a context characterized by a paucity of data [13]. This approach has already been used in patient safety, to explore issues that may not be fully captured by reported data [14,15]. Therefore, this study aims to estimate the frequency of patient safety incidents (PSIs) occurring in inpatient psychiatric facilities according to expert opinion adopting a robust framework and to compare them to the available literature on general healthcare settings.

## 2. Material and Methods

This study was based on a seven-step process, which is shown in Figure 1.

### 2.1. Step 1: Comprehensive Knowledge Synthesis

We adopted a bi-directional strategy to comprehensively gather relevant knowledge on which to develop the questionnaire and understand the emerging results. In detail, the “top-down approach” aimed to identify and use an already existing and acknowledged framework for patient safety, and the “bottom-up approach” to extract evidence on patient safety from the literature. Therefore, this step encompasses two sub-phases, namely “patient safety framework identification” (Step 1A, top-down approach) and “literature review” (Step 1B, bottom-up approach).

#### 2.1.1. Step 1A: Framework Identification

As known by the authors, no specific and comprehensive patient safety frameworks have been specifically developed for the psychiatric setting. Therefore, we adopted the Conceptual Framework for the International Classification for Patient Safety (ICPS) [16], published by the World Health Organization in 2009, which is designed “to be a genuine convergence of international perceptions of the main issues related to patient safety and to facilitate the description, comparison, measurement, monitoring, analysis and interpretation of information to improve patient care”.

Unlike other incident classifications using bottom-up approaches, the ICPS provides a comprehensive, theory-driven taxonomy applicable across diverse healthcare contexts [17], especially valuable in psychiatric settings with potentially under-reported safety incidents.

In detail, we used the “Incident Type” class, which groups incidents based on their common features. This class has 13 different main incident type categories (ITCs), each of which is articulated in additional sub-categories. In the past years, several classifications for PSIs occurring in the psychiatric setting were developed. However, all of them adopted a bottom-up approach [18], in which categories were defined starting from the reported incidents. Although this approach allows the creation of a tailored taxonomy, it is unlikely to provide a comprehensive and theoretically complete classification [19]. This could lead to not including PSIs that are uncommon, perceived as rare, or under-investigated in the specific psychiatric setting [2].

#### 2.1.2. Step 1B: Literature Review

An exploratory literature review was conducted to understand the current state-of-the-art of research about patient safety in general healthcare and in the psychiatric inpatient setting. The results of this review were used as the knowledge base for quality control, more specifically for assessing the inconsistency.

### 2.2. Step 2: Questionnaire Design

The questionnaire consisted of three sections: demographics, ITCs, and PSIs. The complete questionnaire is available in the Appendix A.

In the first section, demographic and job data were collected. Specific information about the involvement of respondents in patient safety (e.g., time dedicated to patient safety-related activities, and official appointments related to patient safety) was gathered. Respondents were asked to estimate the overall rates of PSIs and of healthcare workers who became second victims at least one time in inpatient psychiatric facilities. The second section aimed to estimate the contribution of each ITC to all the PSIs. The 13 different ITCs were proposed to respondents. The third section was built on the ICPS taxonomy and comprised a total of 110 different PSIs. Participants were asked to indicate the percentage of contribution for each PSI to the corresponding ITC.

### 2.3. Step 3: Sample Size

Considering the qualitative descriptive design of our study, the sample size has been established based on the available literature. In detail, our survey aimed to reach saturation, namely “the point at which gathering more data about a theoretical construct reveals no new properties, nor yields any further theoretical insights about the emerging grounded theory” [20]. Despite there not being an univocal consensus on how to determine the sample size [20], it has been estimated that around 25 subjects are enough to reach the meaning saturation in a homogeneous population [21].

### 2.4. Step 4: Expert Identification

Since our questionnaire aimed to gather expert opinions on patient safety, we targeted professionals with an active interest toward patient safety. In detail, we contacted subjects involved in international projects or organizations working on patient safety, namely the International Society for Quality in Health Care (ISQua) [22] (“ISQua—The International Society for Quality in Health Care”, n.d.) and the European Researchers’ Network Working on Second Victim (ERNST) [23]. Such organizations have been chosen considering their aim and the professional profile of their members. ISQua, an international not-for-profit organization dedicated to promoting quality improvement in health care has a network of health professionals spanning over 70 countries, while ERNST brings together various disciplines, backgrounds, and organizations aiming to “facilitate discussion and share scientific knowledge, perspectives, and best practices concerning adverse events in healthcare institutions”.

### 2.5. Step 5: Online Survey

The questionnaire was developed using the Sogolytics online software (https://www.sogolytics.com/; accessed 18 October 2024). All questions and answers were written in English and cross-checked by the research team to ensure their clarity and readability. Moreover, the questionnaire was internally piloted and tested before the deployment. The questionnaire was sent by email to experts, followed by two reminders dispatched at one-week intervals following the initial distribution. Responses were collected anonymously, and participants were asked for informed consent at the beginning of the questionnaire.

### 2.6. Step 6: Data Quality

Data were exported from Sogolytics to an Excel spreadsheet. To ensure data quality, we excluded answers provided by careless responders [24]. In detail, we took in consideration the second section of the questionnaire (ITC) to assess invariability and inconsistency. Both invariability (i.e., consecutive identical responses or identical patterns of response) and inconsistency (i.e., data not matching any expected result based on available evidence) were independently assessed by two researchers (AC and MP), and disagreements were solved after the researcher group discussion. Invariability was assessed for each response, comparing the rates declared for each ITC. Responses containing at least five equal, non-zero rates were therefore excluded. The inconsistency was assessed by comparing questionnaire results with information about patient safety from the general healthcare and psychiatric setting gathered during the conduction of the exploratory literature review. Therefore, responses containing information in sharp contrast with the literature (e.g., reported ten-fold rates for a PSI which is unlikely to occur in the psychiatric setting) were excluded.

### 2.7. Step 7: Data Analysis and Interpretation

We used R 4.1.2 (R Core Team, Vienna, Austria) and Microsoft Excel (Microsoft Corporation, Redmond, WA, USA) for data processing and statistical analysis. Descriptive statistics were used to present the results. In detail, demographic and professional characteristics were presented by frequencies (percentage). For the second and the third sections, results were presented as mean, and the standard deviation (SD) was calculated. It is worth mentioning that, due to the nature of the questionnaire, it was not possible to use Cronbach’s Alpha to calculate the internal consistency [25]. However, we calculated the two-way random, average score intraclass correlation (ICC2K) among the 13 different ITCs and among the PSIs within each ITC for assessing internal consistency. Results were therefore analyzed by adopting a two-level approach. The first level aimed to identify the macro-areas of patient safety in the inpatient psychiatric setting. Therefore, reported relative frequencies of ITCs were analyzed. In the second level, we assessed the relative frequencies of PSIs from the ITCs of which the mean relative frequency was above a predetermined threshold (i.e., 7.5%). The decision to not consider PSIs from the ITCs not reaching the threshold was based on the fact that such answers are likely to be affected by the ratio bias [26], and might not be reliable. Ratio bias is a cognitive phenomenon where humans tend to misinterpret proportional information. It can lead to unreliable estimates, particularly when reporting rare or uncommon events. While the 7.5% threshold was arbitrarily established by the authors, it provides a pragmatic method to focus on more frequent PSIs. This approach allows us to concentrate on the most prominent and potentially actionable patient safety issues while reducing the influence of potential cognitive distortions in expert reporting. Therefore, with an explorative aim, we compared the reported frequencies with the published literature gathered during the literature review phase.

## 3. Results

The survey was deployed in June 2024. A total of 131 respondents from 15 different countries took part in the questionnaire, of whom 39 (29.8%) completed it in its entirety. After the data quality assessment, a total of 33 responses were included.

Demographic information is shown in Table 1 and in Figure 2. The majority of respondents were female (57%), had a management job (39%), and were medical doctors (33%). Age varied from 28 to 72 years old, while the seniority ranged from 5 to 35 years. Forty-two percent of the respondents declared they usually spend most or all of their working time on patient safety-related activities. Twenty respondents (60%) officially held an official appointment related to patient safety.

Overall, the expert estimated that the rate of patients experiencing at least one adverse event during the stay was slightly higher in the inpatient psychiatric setting (mean: 13.91%, median: 13%, SD: 8.71) than the estimation provided by the World Health Organization (10%) [1]. In contrast, respondents pointed out a proportion of healthcare personnel working in the inpatient psychiatric setting who have been a second victim at least one time was estimated lower (mean: 50.55%, median: 60%, SD: 25.25) than the rate reported from the general healthcare setting (60%) [27].

When it comes to the ITC (Table 2 and Figure 3), the predominant categories accounting for the majority of PSIs were Behavior (29.9%), Medication/IV Fluids (9.5%), and Infrastructure/Building/Fixtures (8.6%). Conversely, the least reported ITCs were Blood/Blood Products (2.0%), Oxygen/Gas/Vapor (2.4%), and Healthcare-associated infections (3.4%). The ICC2K among the different ITCs was high (0.98).

In Table 3, we reported the detailed proportion of PSIs from the ITC accounting for 7.5% or more of the total. Overall, the most important ones were Nonexistent/Inadequate Infrastructure/Building/Fixtures (4.9%), Noncompliant/Uncooperative/Obstructive Behavior (4.4%), Damaged/Faulty/Worn Infrastructure/Building/Fixtures (3.8%), Intended Self-Harm/Suicide Behavior (3.4%), and Inconsiderate/Rude/Hostile/Inappropriate Behavior (3.2%).

## 4. Discussion

To the best of the authors’ knowledge, this is the first study aimed at estimating the relative frequency of diverse PSIs in the inpatient psychiatric setting based on expert opinion. Furthermore, it is one of the few studies to adopt the ICPS [34,35], and the first to specifically apply this framework in the psychiatric context. Overall, experts reported a slightly higher rate of patients experiencing at least one PSI in psychiatric inpatient compared to general care. This is in line with the literature, which identified unique patient safety challenges for this setting [2], which is also considered at high risk of different types of PSIs [36]. Interestingly, experts reported a lower rate of SV in the psychiatric setting than in general healthcare [27]. Despite no research assessing SV prevalence having been specifically conducted in this setting, a recent cross-sectional study reported that more than 80% of the nurses suffered from psychological harm after violent episodes in psychiatry [37]. From this point of view, it is reasonable to consider that such a rate could have been underestimated. This could be due to the mental health stigma phenomenon, namely the fact that the stigma and discrimination toward psychiatric and psychological diseases are increased in mental health settings, even by professional staff, leading to a normalization of mental health conditions [38].

Our findings are overall aligned with the existing literature in the field. In this regard, Thibaut et al. conducted a comprehensive systematic review to explore the research landscape and delineate the primary research categories pertaining to patient safety in psychiatric inpatient settings [2]. It is worth mentioning that the main research categories identified in their review, such as interpersonal violence, self-harm, physical environment, and medication safety, are similarly reflected in our results.

When it comes to the ITC, it was possible to compare our results with a retrospective study conducted in two general hospitals [39], which adopted the ICPS for data classification. Notably, experts suggested higher rates for seven ITCs than the one reported from general care (i.e., Medication/IV Fluids, Infrastructure/Building/Fixtures, Clinical Administration, Medical Device/Equipment, Nutrition, Healthcare-Associated Infections, and Blood/Blood Products).

As reported in previous reviews, medication errors are frequent in psychiatric settings, and most such PSIs could be specifically attributed to psychotropic medication. Although the literature identified some specific contributing factors for this setting, the lack of comprehensive research has been pointed out [40,41]. Despite that this is also reflected in the paucity of studies on mitigation strategies for such incidents, it has been suggested that fostering a safety culture and implementing an incident-reporting system could be among the most effective strategies to reduce medication errors [42,43].

Interestingly, experts identified Infrastructure/Building/Fixtures as the fourth main ITC for the psychiatric inpatient setting. To a greater extent than in other settings, the environment of psychiatric inpatients plays a pivotal role in patient safety. For example, it has been shown that physical barriers can exacerbate the occurrence of incidents in inpatient psychiatry [44], as well as room/unit layout, lighting, and noise [4]. This relevance, which is also acknowledged by a recent umbrella review on hospital design [45], is, however, not reflected in the literature, which is scarce [2]. Healthcare administrators and facility managers should prioritize the creation of wide, open-layout spaces with soft furniture, calming colors, and natural lighting while providing patients with elements of control and privacy that might reduce stress and potentially aggressive behaviors. Additionally, clear signage, access to external spaces, and patient-friendly room designs could reduce environmental triggers, ultimately helping to prevent PSIs [4,44,45].

When it comes to clinical administration, available evidence highlights the role of patient transition between different care contexts (i.e., admission, discharge, transfer of care) as a complex process that poses relevant patient safety risks [46]. Therefore, psychiatric institutions should develop context-specific medication protocols detailing the medication review and reconciliation process and should deploy staff training on psychotropic medication management.

This is also acknowledged in psychiatry, where not only this phenomenon is widely studied, but also different interventions to increase safety (such as the implementation of education programs and motivational aftercare planning) have been described [47].

Despite the literature on Medical Device/Equipment in psychiatry being scarce and mainly focused on contingent situations such as the COVID-19 pandemic [48] or medical emergencies [49], available evidence highlights how the equipment available in psychiatric inpatient settings (e.g., restraints, beds) is often difficult to use and also unsuitable for ensuring safety [50]. However, scheduled and effective maintenance, as well as the adoption of standardized devices, could help prevent such incidents [49,50].

Nutrition represents a major issue in psychiatry [51], and inpatient psychiatry has been recognized as an obesogenic environment [52]. While this effect can be partially explained by medication- and patient-related factors, it has also highlighted the role of missing and inadequate food, as well as insufficient mealtime assistance [53].

Previous studies confirmed the high relevance of healthcare-associated infections in the psychiatric setting. Indeed, this setting presents unique challenges, such as close staff-patient contact, communal living environment, and freedom of movement [54]. Moreover, it has been hypothesized that the use of antipsychotic drugs might be associated with an increased risk of infection [55], and additionally, mental health professionals showed a low adherence to infection and prevention control measures [56,57]. However, in line with the other ITCs, evidence about this PSI in the psychiatric setting is particularly limited [2,54]. A periodical review of the prescribed antipsychotic drugs and the implementation of a structured infection prevention and control system could support the reduction in healthcare-associated infections [54,58].

Interestingly, experts reported a high rate of Blood/Blood Products-related incidents. Even if the literature on this topic is scarce, it has been suggested that transfusion-related PSIs might have a higher incidence in specific populations such as children and psychiatric patients [59], and errors in blood transfusion are noted to occur frequently in psychiatric centers, where the compact nature of facilities may hinder proper monitoring and management [60].

Results from our study have relevant implications for clinical practice and for patient safety in the psychiatric inpatient setting. Some ITCs were considered by the experts more frequent in the psychiatric inpatient setting than in general care [39] (i.e., Infrastructure/Building/Fixture). Our study, which was based on capturing experts’ knowledge rather than collecting available evidence, aimed to overcome the limitations of the literature review. Indeed, knowledge synthesis is intrinsically suffering from the lack of studies on a certain area or topic. In this regard, the review published by Thibaut et al. [2] shows two significant issues: first, there is a lack of evidence regarding patient safety in psychiatric settings; second, in the specific safety areas where our study identified a higher proportion of incidents, only anecdotal evidence is currently available.

This concerning lack of evidence might be attributed to different factors. Historically, psychiatric care has been marginalized within broader healthcare research, with lesser attention given to quality compared to other medical areas [61]. The stigma surrounding mental health, together with the traditional focus on clinical outcomes rather than safety processes, could have hampered a systematic analysis of PSIs [38]. Moreover, it has been observed that research on patient safety in psychiatry has traditionally been addressed with a disproportionate focus on the prevention of violence, self-harm, and suicide, probably missing a more comprehensive approach [62].

Our deductive approach, which started from a general framework (the ICPS) rather than from primary studies, allowed us to identify underinvestigated areas. Moreover, it is noteworthy that the few pieces of evidence available from such ITCs suggest the relevance, both in terms of frequency and severity, of the related PSI. For example, psychotropic medications, one of the most used medications in the inpatient psychiatric setting, are associated with a higher risk of severe incidents. However, several studies highlighted the alarming lack of evidence on medication safety in psychiatry. Similarly, despite the psychiatric setting having been identified as a setting at risk of healthcare-associated infections, there are only a few studies investigating such events as patient safety issues.

This discrepancy could be understood considering the intrinsic limitations of incident reporting, on which most of the available evidence is based. Despite such systems having been proven to be overall effective in monitoring and reducing the occurrence of PSIs [63], they suffer from underreporting, especially regarding rare events or not fully perceived as PSIs [64]. Moreover, each incident-reporting system has a tailored taxonomy, which usually reflects the characteristics of PSIs expected from a specific setting [7]. While the adoption of a context-specific taxonomy is useful to facilitate the reporting process, it might reduce the reporting of uncommon incidents [64]. From this point of view, expert opinion might be useful not only to identify under-researched areas but also to identify priorities in developing and implementing patient safety initiatives. For example, experts identified the same rate of behavior-related incidents in psychiatric inpatients compared to general care. If from one side this is the most relevant ITC, accounting for one-third of the total, from the other side, it is also the category with the most research and interventions available [2]. Our results, therefore, not only suggested potentially relevant patient safety areas in which research is limited but also in which there is a need for the development and deployment of patient safety measures specifically designed for inpatient psychiatry. Moreover, the approach we used to develop and conduct this study could be easily translated to other specific contexts, allowing a deep understanding of patient safety.

Our findings suggest some implications for both clinical practice and future research. Clinically, healthcare administrators and professionals should focus on the identified PSI categories (e.g., behavior-related incidents, medication, infrastructure) by developing and implementing specifically tailored interventions. Recognizing the unique challenges emerging from the psychiatric inpatient setting environment is pivotal to developing detailed safety protocols addressing PSIs in these settings. With regard to future research, our study underscores the need for exploring under-investigated patient safety domains, potentially adopting methodologies different from traditional incident-reporting systems. Additionally, fostering interdisciplinary collaborations can facilitate a more comprehensive understanding of safety aspects, ultimately fostering evidence-based strategies.

### Limitations

Our study presents some limitations. Therefore, considering all the below-mentioned issues, we recommend caution in interpreting our results. First, the limited sample size might affect the generalizability and the statistical power of our findings. The restricted number of participants could potentially introduce sampling bias and limit a comprehensive representation of PSIs in inpatient psychiatric settings. Moreover, since we asked for an estimation based on the experience gained during the entire career, we can not exclude the occurrence of the recall bias. Therefore, interpretations should be approached with caution, recognizing that our results may not fully capture the entire spectrum of safety challenges in psychiatric care and might not be generalizable to all psychiatric settings. Second, it should be noted that for an ITC (i.e., Oxygen/Gas/Vapor) and nine PSIs (i.e., Problem with Substance Use/Abuse, Harassment, Sexual Assault, Death Threat, Blunt Force, Other Mechanical Force, Exposure to Chemical or Other Substance, Other Specified Mechanism of Injury, Exposure to/Effect of Weather, Natural Disaster, or Other Force of Nature) it was not possible to identify an adequate comparison in the literature. Indeed, while some such items have not been yet investigated at all (e.g., Exposure to Chemical or Other Substance) others have been studied without comparing their frequency to the total of reported PSIs (e.g., Sexual Assault [65]). Third, seven respondents considered the provided list of PSIs as non-exhaustive. In detail, the PSIs that were considered missing were: technical issues, data breaches, use of medical devices, omitted/delayed/wrong diagnosis, inappropriate patient restraint, non-compliance with medications, and mechanical restraint-related injuries. Despite our questions accurately representing the ICPS taxonomy, this framework, even if it has been designed to be easily adapted to specific contexts, could have missed some specific aspects that were deemed important for a psychiatric setting. Fourth, since the questionnaire was anonymous, we were unable to evaluate the effective expertise of the respondents. However, since the survey was sent to members of two internationally relevant organizations working in patient safety, we are confident that the results represent the position of most patient safety professionals. This is also supported by the high ICC2K values, which showed a high consistency between experts’ opinions. However, we recommend caution in interpreting our results, since self-reported information intrinsically suffers from several biases (e.g., recall, confirmation, response, cognitive, and cultural biases). Moreover, it should be noted that we matched expert opinions with primary data gathered from diverse primary studies. While this comparison might provide a general overview of the differences in patient safety across different care settings, it is important to acknowledge that the difference in percentages derived from such comparisons must not be considered highly precise. Diverse study designs, data collection techniques, and contextual differences make imperative an approximate interpretation of the results. Subsequently, our findings should be viewed as a broad approximation rather than an exact quantification of PSIs. Future research conducted adopting similar approaches should try to expand the sample size by including a wider range of institutions and healthcare professionals, to provide a more representative analysis.

## 5. Conclusions

Our study showed a novel approach to understanding patient safety in the inpatient psychiatric setting, considering experts’ opinions within the ICPS. While the rate of PSIs in psychiatric inpatient settings is higher than in general healthcare, the contribution of ITCs to the overall is slightly different. Behavior-related incidents, medication errors, and infrastructure issues emerged as the most frequent ITCs. Our findings could support clinicians and healthcare leaders in prioritizing safety improvement efforts in the psychiatric inpatient setting. Moreover, future research should focus on developing and evaluating specific safety interventions tailored to the unique challenges of this context, particularly in areas where current evidence is limited.

## Figures and Tables

**Figure 1 behavsci-14-01116-f001:**
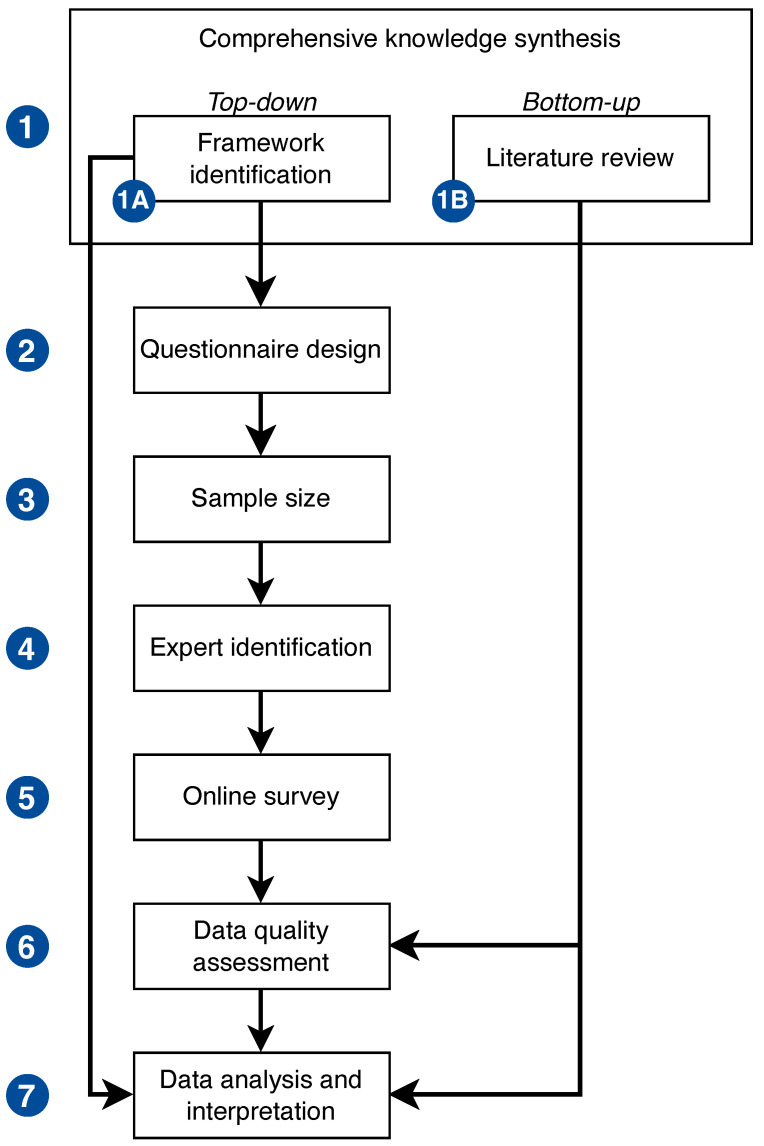
The seven-step process.

**Figure 2 behavsci-14-01116-f002:**
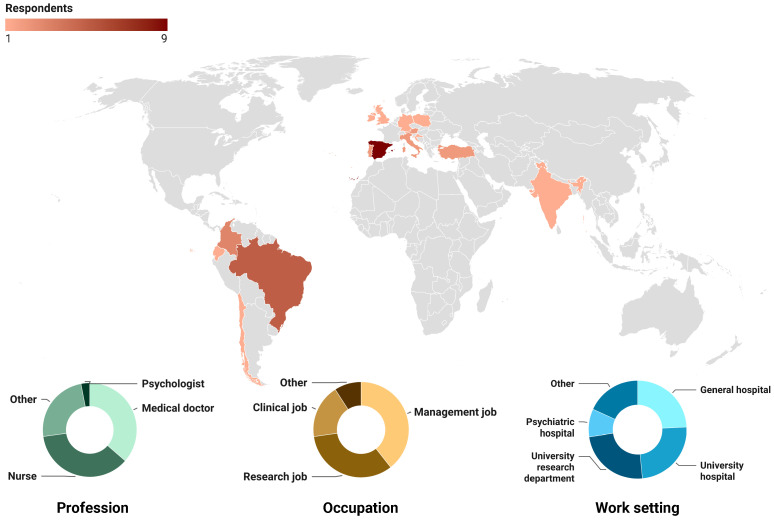
Visual representation of respondent demographics.

**Figure 3 behavsci-14-01116-f003:**
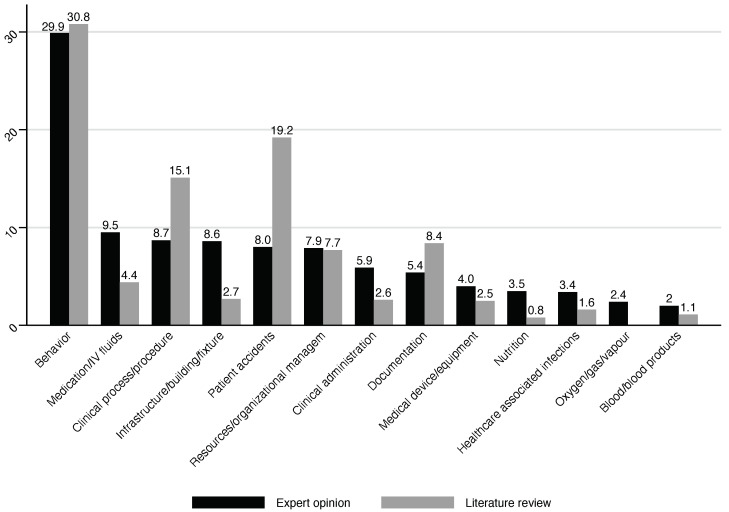
Comparison of ITC rates (as percentage) of psychiatric inpatient setting (Expert opinion) and general care (Literature review).

**Table 1 behavsci-14-01116-t001:** Demographic information of respondents.

Gender (n, %)	
Female	23 (69.7)
Male	10 (30.3)
Other	0 (0.0)
**Country (n, %)**	
Austria	2 (6.1)
Brazil	5 (15.2)
Chile	1 (3.0)
Colombia	3 (9.1)
Croatia	1 (3.0)
Ecuador	1 (3.0)
Germany	1 (3.0)
India	1 (3.0)
Ireland	1 (3.0)
Italy	2 (6.1)
Portugal	2 (6.1)
Poland	1 (3.0)
Spain	9 (27.3)
Turkey	2 (6.1)
United Kingdom	1 (3.0)
**Work setting (n, %)**	
General hospital	8 (24.2)
Non-university research structure	2 (6.1)
Non-clinical healthcare institutions and bodies	1 (3.0)
Primary care	1 (3.0)
Psychiatric hospital	3 (9.1)
University hospital	8 (24.2)
University research department	8 (24.2)
Other	2 (6.1)
**Occupation (n, %)**	
Clinical job	6 (18.2)
Management job	13 (39.4)
Research job	11 (33.3)
Other	3 (9.1)
**Profession (n, %)**	
Medical doctor	12 (36.4)
of which psychiatrist	2 (6.1)
Nurse	12 (36.4)
of which psychiatric nurse	0 (0)
Psychologist	1 (3.0)
Other	8 (24.2)
**Working time spent on patient safety activities (n, %)**	
Full time	7 (21.2)
Most of my working time	7 (21.2)
Half time	5 (15.2)
Less than half	10 (30.3)
No time	4 (12.1)
Official patient safety appointment (n, %)	20 (60.7)
Considers PSI list not exhaustive (n, %)	7 (21.2)

**Table 2 behavsci-14-01116-t002:** ITC rates reported by respondents.

Incident Type Category	Expert Opinion (Mean %, SD, ICC2K)	Literature Review (%)
Behavior	29.9 (17.9, 0.91)	30.8
Medication/IV Fluids	9.5 (11.6, 0.90)	4.4
Clinical Process/Procedure	8.7 (6.6, 0.93)	15.1
Infrastructure/Building/Fixtures	8.6 (7.2, 1.00)	2.7
Patient Accidents	8.0 (8.1, 0.95)	19.2
Resources/Organizational Management	7.9 (5.7, 0.89)	7.7
Clinical Administration	5.9 (5.4, 0.89)	2.6
Documentation	5.4 (3.6, 0.66)	8.4
Medical Device/Equipment	4.0 (3.5, 0.86)	2.5
Nutrition	3.5 (2.9, 0.90)	0.8
Healthcare-Associated Infections	3.4 (2.9, 0.72)	1.6
Oxygen/Gas/Vapor	2.4 (1.7, 0.93)	-
Blood/Blood Products	2.0 (1.1, 0.89)	1.1

**Table 3 behavsci-14-01116-t003:** Patient safety incident rates reported by respondents.

Patient Safety Incident	Expert Opinion (%)	Literature Review (%)
**Patient behavior**		
Noncompliant/Uncooperative/Obstructive	4.4	3.0 [28]
Inconsiderate/Rude/Hostile/Inappropriate	3.2	0.3 [28]
Risky/Reckless/Dangerous	3.1	1.5 [28]
Problem with Substance Use/Abuse	3.2	-
Harassment	1.4	-
Discrimination/Prejudice	1.2	0.5 [28]
Wandering/Absconding	1.4	0.5 [28]
Intended Self-Harm/Suicide	3.4	0.9 [29]
Verbal Aggression	2.9	0.2 [28]
Physical Assault	2.0	0.2 [28]
Sexual Assault	0.9	-
Aggression Toward an Inanimate Object	1.7	0.2 [28]
Death Threat	1.1	-
**Infrastructure/Building/Fixtures**		
Nonexistent/Inadequate	4.9	0 [30]
Damaged/Faulty/Worn	3.8	0 [30]
Medication/IV Fluids		
Wrong Patient	0.5	0.8 [31]
Wrong Drug	1.1	1.2 [29]
Wrong Dose/Strength of Frequency	1.5	2.1 [32]
Wrong Formulation/Presentation	0.6	2.1 [32]
Wrong Route	0.5	0.5 [31]
Wrong Quantity	0.7	3.4 [31]
Wrong Dispensing Label/Instruction	0.5	0.2 [33]
Contraindication	0.7	0.6 [31]
Wrong Storage	0.5	1.0 [32]
Omitted Medicine or Dose	1.0	0.3 [29]
Expired Medicine	0.3	0.1 [33]
Adverse Drug Reaction	0.9	2.5 [29]
**Patient Accidents**		
Blunt Force	0.7	-
Piercing/Penetrating Force	0.7	0.9 [34]
Other Mechanical Force	0.7	-
Thermal Mechanism	1.3	0.3 [29]
Threat to Breathing	1.2	0.9 [34]
Exposure to Chemical or Other Substance	0.6	-
Other Specified Mechanism of Injury	0.5	-
Exposure to/Effect of wWather, Natural Disaster, or Other Force of Nature	0.7	-
Falls	2.2	11.3 [29]
**Resources/Organizational management**		
Matching of Workload Management	1.7	4.0 [34]
Bed/Service Availability/Adequacy	1.3	0.3 [28]
Human Resource/Staff Availability/Adequacy	1.9	2.8 [28]
Organization of Teams/People	1.5	0.0 [28]
Protocols/Policy/Procedure/Guideline Availability/Adequacy	1.5	0.5 [28]

## Data Availability

Dataset available on request from the authors.

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
