# Peer review of "Patient Safety Incidents in Inpatient Psychiatric Settings: An Expert Opinion Survey"

_behavsci, 2024, doi:10.3390/bs14111116_

Round 1

Reviewer 1 Report

Comments and Suggestions for Authors

This is a very strong article about a critically important topic. Relying upon expert knowledge is an important dimension to understand and be able  to respond to issues of patient safety.

The Abstract is very clear and well-written.

The literature review is good, utilizing both older and very recent references. The study uses very recent data from June 2024, another strength of this work.

I do suggest that the authors add why they think patient safety has been overlooked over time.

Also, I suggest the authors add a review of the consequences for patients and staff when safety issues are overlooked. Patient and staff safety must be given the highest priority, especially on psychiatric units where behavior problems are not uncommon. It cannot be a therapeutic environment when there is fear of patients by staff and fear of staff by patients. (I know of one recent case in the US where a patient in psychosis received three assault and battery charges against staff in the local district court for behavior problems associated with psychosis. The family has been unable to move the patient to a higher level of care in a different psychiatric unit. It is absurd to think the hospital is handling this situation well. Of course, there is also the question of whether criminal charges can or should be appropriately leveled against severely mentally ill patients.)

Line #31 remove the letter "a" before "patient safety issues."

Figure 1 shows the seven-step process very clearly. The description of the process is very good. I am not a research methodologist/statistician and I hope another reviewer can be helpful here. Good use of grounded theory.

Figure 2 looks good and gives a good picture of the demographics. The  caption should be corrected to either "respondents' demographics" or "respondent  demographics."

The Tables look good to  me.

The Discussion section nicely highlights the importance of this work, with the first paragraph in this section especially strong.

Perhaps the most important  thing about this work is the light it shines on  the need to further study and address patient/staff safety, medication  errors, and use of blood products.

I hope the authors will continue to address these problems in the future work.

Author Response

Dear Reviewer, we wish to thank you for the time spent on revising our manuscript and for the kind report. Please find below our answers to your comments:

This is a very strong article about a critically important topic. Relying upon expert knowledge is an important dimension to understand and be able  to respond to issues of patient safety. The Abstract is very clear and well-written. The literature review is good, utilizing both older and very recent references. The study uses very recent data from June 2024, another strength of this work.

I do suggest that the authors add why they think patient safety has been overlooked over time.

We added our speculations, please see lines 274-281.

Also, I suggest the authors add a review of the consequences for patients and staff when safety issues are overlooked. Patient and staff safety must be given the highest priority, especially on psychiatric units where behavior problems are not uncommon. It cannot be a therapeutic environment when there is fear of patients by staff and fear of staff by patients. (I know of one recent case in the US where a patient in psychosis received three assault and battery charges against staff in the local district court for behavior problems associated with psychosis. The family has been unable to move the patient to a higher level of care in a different psychiatric unit. It is absurd to think the hospital is handling this situation well. Of course, there is also the question of whether criminal charges can or should be appropriately leveled against severely mentally ill patients.)

We added this information in the introduction, to strengthen the rationale for the conduction of our study.

Line #31 remove the letter "a" before "patient safety issues."

We corrected this error.

Figure 1 shows the seven-step process very clearly. The description of the process is very good. I am not a research methodologist/statistician and I hope another reviewer can be helpful here. Good use of grounded theory.

Thank you for this comment. Since we adopted this newly developed method, we are glad that Figure 1 can support the readers to understand our approach.

Figure 2 looks good and gives a good picture of the demographics. The  caption should be corrected to either "respondents' demographics" or "respondent  demographics."

We corrected the caption according to this suggestion.

The Tables look good to  me. The Discussion section nicely highlights the importance of this work, with the first paragraph in this section especially strong. Perhaps the most important  thing about this work is the light it shines on  the need to further study and address patient/staff safety, medication  errors, and use of blood products. I hope the authors will continue to address these problems in the future work.

Reviewer 2 Report

Comments and Suggestions for Authors

Thank you for your insightful manuscript on patient safety incidents (PSI) in psychiatric inpatient settings. Below, few comments you might consider \clarify

Clearly acknowledge in the manuscript that the small sample size limits quantitative generalizability and statistical power. Specify that, while the sample size allowed for qualitative insights and thematic saturation, it may not fully capture the diversity of PSI experiences across psychiatric care settings

Expand the Discussion section to explain how the limited sample size may influence interpretations and why caution should be applied when generalizing findings to all psychiatric settings. Adding a recommendation for future studies to include a larger sample size would reinforce this.

acknowledgment\Discussion: The exclusion of less frequently reported incident types (e.g., medical device incidents, data breaches) could result in the underestimation of certain impactful PSI categories in psychiatric care.

Provide concrete recommendations in the Discussion section for mitigating prevalent PSIs. This could include protocols for handling behavior-related incidents, enhancing environmental safety measures, and improving medication management. Providing actionable solutions tailored to psychiatric care would increase the manuscript’s relevance for clinical settings.

  •  In Table 2, Figure 3, and the text, there are minor rounding inconsistencies in PSI category percentages (e.g., behavior incidents at 29.9% and others). Ensuring uniformity across these elements will provide readers with a clearer view of PSI frequencies and minimize any confusion. Consider a single decimal point rounding rule or exact percentages consistently applied across the text and visuals.

  •  In Table 3, some categories reported by the expert panel (e.g., “substance use” and “harassment”) lack a corresponding literature benchmark. Providing additional context in the text regarding why certain psychiatric-specific incidents do not appear in general healthcare literature comparisons would enhance reader understanding.

  • Looking forward to read the revised manuscript, best wishes

Author Response

We wish to thank the reviewer for the effort in revising our manuscript and the detailed report. Please find below our answers. Changes in the manuscript are highlighted in red.

Thank you for your insightful manuscript on patient safety incidents (PSI) in psychiatric inpatient settings. Below, few comments you might consider\clarify.

Clearly acknowledge in the manuscript that the small sample size limits quantitative generalizability and statistical power. Specify that, while the sample size allowed for qualitative insights and thematic saturation, it may not fully capture the diversity of PSI experiences across psychiatric care settings. Expand the Discussion section to explain how the limited sample size may influence interpretations and why caution should be applied when generalizing findings to all psychiatric settings. Adding a recommendation for future studies to include a larger sample size would reinforce this.

We agree with this suggestion, and we acknowledged such issues in the limitation paragraph (at the end of the discussion). Moreover, we moved limitation in a dedicated sub-paragraph to facilitate the readers to identify them.

Acknowledgment\Discussion: The exclusion of less frequently reported incident types (e.g., medical device incidents, data breaches) could result in the underestimation of certain impactful PSI categories in psychiatric care.

We addressed this issue in the discussion. We decided to focus on the most reported incidents to keep readers' attention high and to have a discussion of appropriate length. However, we agree that also less reported incident type could pose serious challenges in psychiatric care. 

Provide concrete recommendations in the Discussion section for mitigating prevalent PSIs. This could include protocols for handling behavior-related incidents, enhancing environmental safety measures, and improving medication management. Providing actionable solutions tailored to psychiatric care would increase the manuscript’s relevance for clinical settings.

We agree with the reviewer about increasing the relevance of the manuscript. We added recommendations in the discussion. 

In Table 2, Figure 3, and the text, there are minor rounding inconsistencies in PSI category percentages (e.g., behavior incidents at 29.9% and others). Ensuring uniformity across these elements will provide readers with a clearer view of PSI frequencies and minimize any confusion. Consider a single decimal point rounding rule or exact percentages consistently applied across the text and visuals.

We fixed such inconsistencies in tables and figures, using single decimal point.

In Table 3, some categories reported by the expert panel (e.g., “substance use” and “harassment”) lack a corresponding literature benchmark. Providing additional context in the text regarding why certain psychiatric-specific incidents do not appear in general healthcare literature comparisons would enhance reader understanding.

Unfortunately, despite our accurate literature review, we did not find suitable benchmarks. Indeed, despite some of such ITC/PSI were investigated by few studies, the frequency was reported using a different format which could have not been used in our study (for example: the number of harassment without specifying the total of PSI reported, or the percentage of harassment events out of behavioral -and not total- PSI). We added this explanation in the limitations (expanding the second point).

Looking forward to read the revised manuscript, best wishes

Round 2

Reviewer 2 Report

Comments and Suggestions for Authors

Thank you for addressing the comments

Author Response

We wish to thank the Reviewer for the effort in assessing our manuscript.